# Reliability and Validity of the Treatment Satisfaction with Medicines Questionnaire (SATMED-Q) in Persons with Arterial Hypertension

**DOI:** 10.3390/ijerph18063212

**Published:** 2021-03-19

**Authors:** Jesús López-Torres López, Joseba Rabanales-Sotos, María Rosa López-Torres Hidalgo, Rosa María Milián García, Consuelo López Martínez, Gemma Blázquez Abellán

**Affiliations:** 1Community Pharmacy, 02005 Albacete, Spain; jlopeztl91@gmail.com (J.L.-T.L.); miliangarciarosa@gmail.com (R.M.M.G.); consuelolopez@redfarma.org (C.L.M.); 2Department of Nursing, Physiotherapy and Occupational Therapy, Albacete Nursing Faculty, University of Castile-La Mancha, 02071 Albacete, Spain; 3Department of Medical Sciences, Albacete Faculty of Pharmacy, University of Castile-La Mancha, 02008 Albacete, Spain; mrosa.lopeztorres@uclm.es (M.R.L.-T.H.); gemma.blazquez@uclm.es (G.B.A.)

**Keywords:** hypertension, patient satisfaction, community pharmacy services

## Abstract

Objective: To evaluate the reliability and validity of the Treatment Satisfaction with Medicines Questionnaire (SATMED-Q) in persons with arterial hypertension undergoing pharmacological treatment, along with its convergent validity with degree of control of blood pressure levels, therapeutic adherence, and tolerability of antihypertensive drugs. Methods: Observational cross-sectional study conducted on a sample of 484 persons. Treatment satisfaction was evaluated with the SATMED-Q, an instrument consisting of 17 items with six dimensions. Other variables were blood pressure, antihypertensive drugs, adverse effects, therapeutic adherence, and participants’ characteristics. Results: Cronbach’s alpha was 0.916. Factor analysis revealed six factors that could account for 89.97% of total variance. The test–retest reliability analysis yielded an intraclass correlation coefficient of 0.910 (95% CI = 0.806–0.959). In a possible range of 0 through 100 points, participant satisfaction with treatment ranged from 38.2 to 100 (mean 79.9 (SD = 12.9; 95% CI = 78.8–81.0); median 80.9). SATMED-Q scores were higher among persons who reported experiencing no adverse effects (82.5 ± 11.6 SD vs. 68.7 ± 11.9 SD; *p* < 0.001). Satisfaction levels were significantly lower among subjects not complying with the treatment (73.2 ± 12.9 vs. 82.1 ± 12.1; *p* < 0.001), and significantly higher among those presenting with controlled blood pressure levels (82.1 ± 12.1 SD vs. 77.5 ± 13.3 SD; *p* < 0.001). Conclusions: The SATMED-Q showed high internal consistency and good stability in the reliability analysis. It is an appropriate instrument for evaluating satisfaction with antihypertensive treatment, both in routine clinical practice and in community pharmacy or clinical research settings.

## 1. Introduction

Treatment satisfaction is defined as “the individual’s rating of important attributes of the process and outcomes of his/her treatment experience” [1]. Patient satisfaction with the treatments they receive is a research area with great potential for providing outcome measures for clinical trials and chronic disease management, whether at physician’s offices or community pharmacies. As has been shown by trials on patients with chronic diseases, patient satisfaction can be more sensitive to change than health-related quality of life [2]. 

Patient satisfaction with treatment predicts its continuity, correct use of medication, and therapeutic adherence to the treatment regimen [3], though there are also other important aspects, such as the quality of healthcare and information about the treatment given by the physician, pharmacist, or other health professionals. In contrast, patient dissatisfaction with treatment can compromise the clinical effectiveness and efficiency of healthcare, including pharmaceutical care. Patients who perceive their treatment to be ineffective, who experience side effects, or who consider that its administration poses certain drawbacks, are less likely to comply with therapeutic recommendations. 

Arterial hypertension (AHT) affects approximately 40% of all adults in developed countries and is considered the most prevalent “controllable” disease in the world [4]. Although it is believed that all pharmacological groups reduce blood pressure levels to a fairly similar degree [5], there a varying situations which render the use of one drug or another advisable, such as combinations of other cardiovascular risk factors, existence of a specific comorbidity that formally indicates or contraindicates the use of a given drug, involvement of target organs, possibility of drug interactions, and socio-economic factors, etc. The choice of drug should thus be individualized in order to ensure integrated treatment of each patient’s cardiovascular risk [6]. Optimal blood pressure control is often difficult, and indeed frequently requires a combination of two or more drugs. Recent years have witnessed a spectacular increase in the number of clinical trials conducted on hypertensive patients, which has made it possible to take an evidence-based approach to AHT treatment, setting the most appropriate blood pressure levels to prevent cardiovascular disease, and being able to choose the most suitable drugs from a large therapeutic arsenal. Furthermore, patients’ decisions to continue, modify, or suspend medical treatments are influenced by different variables, including the desire to participate in the taking of decisions concerning their treatment, health status, prior experiences with treatments, beliefs about efficacy, or adverse effects, etc. [7].

It is important to have patients’ opinions about their medication, and one way of achieving this is to evaluate perceived outcomes objectively. Such patient-reported outcomes should include an evaluation of the treatment received without any interpretation of their responses by the physician, pharmacist, or other health professional, and to ensure this, properly validated instruments are needed. Most treatment-satisfaction studies have used disease-specific questionnaires but for a more generalized use in daily practice, generic questionnaires would seem to be more useful, such as the Treatment Satisfaction Questionnaire for Medication (TSQM) or Treatment Satisfaction with Medicines Questionnaire (SATMED-Q). The latter was designed to be used on chronic patients undergoing pharmacological treatment for any disease. This questionnaire is considered appropriate for use in clinical practice and has been used in recent years to evaluate treatment satisfaction in different clinical situations, ranging from patients with insomnia who receive hypnotic treatment [8], to use of topical agents for rosacea management [9], recipients of solid organ transplants [10], and patients with neuropathic pain [11].

Accordingly, the aim of this study was to evaluate the reliability and validity of the SATMED-Q in persons with arterial hypertension undergoing pharmacological treatment, along with its convergent validity with degree of control of blood pressure levels, therapeutic adherence, and tolerability of antihypertensive drugs.

## 2. Materials and Methods

### 2.1. Study Design and Participants

We conducted an observational cross-sectional study, in which a sample of subjects with AHT who were taking antihypertensive medication, were recruited at two community pharmacies in the city of Albacete (southeastern Spain) and evaluated by personal interview. The study design is shown in Figure 1, and the characteristics of the study subjects are described in greater detail in a previously published paper [12], targeted at evaluating the level of satisfaction of persons with AHT who receive antihypertensive treatment. The inclusion criteria were defined as any person aged over 18 years who was receiving antihypertensive treatment under medical prescription, and who agreed to participate in the study once he/she had been informed of its aims. The following were excluded: subjects with low intellectual performance, due to their presenting with cognitive impairment or severe sensory deficits capable of hindering collaboration in the study; and those who refused to participate.

Assuming a 95% confidence level, a precision of ±4.5%, and an expected indeterminate proportion of persons satisfied with their treatment (*p* = 0.50), an initial sample size of 475 subjects was calculated. Using an expected non-response rate of 20% for study purposes, we applied the formula “Adjusted no. of subjects = No. of subjects (1/(1–expected proportion of losses))” to obtain a final sample size of 593. Consecutive, non-probabilistic sampling was then performed until the envisaged number of subjects had been reached at community pharmacies. Interviews were conducted by four pharmacists across the period October 2017 through December 2018. 

### 2.2. Ethical Considerations

The study protocol received official approval from the Clinical Research Ethics Committee of the Albacete University Teaching Hospital Complex (Spain) on 25 July 2017. This study complied throughout with the ethical principles of voluntariness in participation, guaranteed anonymity for all data furnished by participants, and exclusive restriction of data to the proposed study.

### 2.3. Measures

A data-collection sheet was designed to record the study variables. These included satisfaction with antihypertensive treatment, as evaluated by the SATMED-Q [13], an instrument consisting of 17 items with six dimensions, i.e., efficacy of treatment, ease of use, impact on daily activities, medical care, overall satisfaction, and undesirable side effects. The score range is 0 through 68 points, though it can be transformed into a scale of 0 to 100. Prior to data-collection, authorization to use the SATMED-Q was obtained from the Mapi Research Trust. Other study variables were: blood pressure levels (2 measurements made with an Omron HBP-1300 blood pressure monitor, after a minimum five-minute rest period, with the subject in a seated position, his/her spine firmly supported by the back of the chair and arm in a semi-flexed position at the level of the heart); antihypertensive drugs used (type of drug by subgroup belonging to group C of the Anatomical Therapeutic Chemical classification); possible reported adverse effects related with the antihypertensive medication; therapeutic adherence as evaluated by the Medication Adherence Questionnaire (MAQ) [14]; and participants’ characteristics (age, sex, and educational level).

### 2.4. Statistical Analysis

After the participants’ responses had been entered into a database, processed, and analyzed, a description of the study subjects was drawn up. Prior to evaluating the validity of the SATMED-Q, its feasibility was determined by reference to the percentage of unanswered questions. The first aspect of validity analyzed was the ceiling and floor effect of the items, defined as the percentage of subjects with maximum and minimum responses, respectively. To establish that the questions were related with the total score and that their scoring followed the same direction as the full scale, the linear correlation coefficient was applied and its size, positive sign, and degree of significance noted. Question-total correlation was determined to evaluate the homogeneity index, with correlations below 0.4 being deemed to reflect a lack of relationship between the question concerned and the remaining questions in the questionnaire. The assumption of the SATMED-Q’s internal coherence, or variability of the set of items which the sum of the scores is capable of measuring, was evaluated with Cronbach’s alpha (weighted mean of correlations between variables or items which form part of the questionnaire). 

To ensure adequate construct validity, scale content was analyzed qualitatively, thereby confirming that its content was concordant with the theoretical concept of treatment satisfaction. Factor analysis was used to examine the underlying and fundamental dimensions, with extraction of factors by the maximum likelihood method and varimax orthogonal rotation, in order to establish the various aspects envisaged in the questionnaire. The “loading” or correlation coefficient of each of the items in these factors was evaluated, and their appropriateness ascertained with the Kaiser–Meyer–Olkin test (comparison of magnitudes of partial correlation coefficients) and Bartlett’s test of sphericity (determination of the existence of intercorrelations in the matrix). The established methods used for validating the criterion of questionnaires or measures require the latter to be compared against a gold standard. However, in the case of evaluation of patient satisfaction with antihypertensive treatment, there is no instrument that can be used as such, not even a validated measure, so that the only possible course is to evaluate its relationship with other variables with which it is expected to be related, such as clinical effectiveness (degree of blood pressure control), absence of adverse effects, and good therapeutic adherence. Convergent validity was determined using the *t*-test of comparison of means, and the Spearman correlation coefficient with respect to measurements conceptually related with the construct evaluated (MAQ, presence of adverse effects, and blood pressure levels).

Test–retest stability or reliability was evaluated in a subsample of 25 patients, using the intraclass correlation coefficient between the measurement recorded at baseline and that made several days thereafter. This coefficient, based on variance analysis, is appropriate for showing the changes in mean values, as well as the correlation between the different measures.

## 3. Results

A total of 484 subjects were evaluated, corresponding to an 81.6% response rate. Most persons who refused to participate did so because they did not have the time to undergo the interview, and in a small number of cases because of health reasons. The proportion of women was 56.2% and participants’ mean age was 67.8 years (SD = 11.9). Most of the study subjects had a low educational level, with 18.6% being illiterate or only able to read and write, and 50.4% having solely received primary education. The proportion of subjects who lived alone was 10.5%, while that of subjects who were institutionalized was 2.1%. In terms of occupational activity, 25.4% were unskilled workers, and 26.0% of cases were either unemployed or pensioners.

The most frequent antihypertensive drugs used were angiotensin-converting enzyme inhibitors (ACE inhibitors) (19.0%), angiotensin II receptor antagonists (ARA-II) (9.5%), and diuretics (D) in combination with ARA-II (8.3%). The proportion of participants who reported one or more adverse effects linked to their antihypertensive medication was 19.0% (92 subjects), with the most frequent side effects being dizziness, polyuria, and lower limb edema. Adequate therapeutic adherence with the antihypertensive drug regimen, as rated by the MAQ, was 74.8% (95% CI = 70.8–78.8). The sample registered a mean systolic blood pressure value of 138.9 mmHg (SD = 14.8) and a mean diastolic blood pressure value of 79.9 mmHg (SD = 10.6), with 48.3% of participants registering figures of 140/90 mmHg or higher (95% CI = 43.8–52.9). 

The questionnaire was answered by 484 subjects, all of whom responded to all the items, and of these, 7.2% recorded a maximum response (100 points). There were no cases of minimum response (0 points), with the minimum value obtained being 38.2 points. The homogeneity index, construed as the correlation between individual question scores and the total score, is shown in Table 1 (minimum value: 0.437; maximum value: 0.800). Cronbach’s alpha, or the weighted mean of the correlations between all items forming part of the scale, was 0.916.

The underlying dimensions of the SATMED-Q were examined by factor analysis, with extraction of factors by principal component analysis and posterior rotation by the varimax method (Table 2). The Kaiser–Meyer–Olkin measure of sampling adequacy or suitability of data was 0.854, while Bartlett’s sphericity test, targeted at evaluating the applicability of factor analysis, yielded a figure of 8.644.04 with 136 degrees of freedom (*p* < 0.001), thus making it possible to conclude that there were significant correlations between attributes. 

Factor analysis revealed that there were six factors capable of accounting for 89.97% of total variance (the only ones whose eigenvalues exceeded a value of 1) (Table 3). Table 4 shows the variance explained in each factor in comparison with the results of the original study. The items with greatest saturation were included in each factor, and their content was interpreted to configure each dimension. Table 5 shows the loadings obtained by the items in the factor analysis.

The test–retest reliability analysis, targeted at ascertaining the stability of the measurements in a subsample of patients (mean age 64.8 years (SD = 9.1); 64.0% women) who were evaluated at the end of one week, yielded an intraclass correlation coefficient of 0.910 (95% CI = 0.806–0.959). 

In a possible range of 0 through 100 points, participant satisfaction with treatment ranged from 38.2 to 100 (mean 79.9 (SD = 12.9; 95% CI = 78.8–81.0); median 80.9). SATMED-Q scores proved to be higher in persons who reported experiencing no adverse effects as a consequence of the antihypertensive medication (82.5 ± 11.6 SD vs. 68.7 ± 11.9 SD; *p* < 0.001). The level of satisfaction was significantly lower in persons not complying with the treatment (73.2 ± 12.9 vs. 82.1 ± 12.1; *p* < 0.001). Analysis of blood pressure levels showed that there was a very weak negative correlation, albeit statistically significant, between satisfaction scores and pressure figures, both systolic (r =−0.138; *p* = 0.002) and diastolic (r = −0.178; *p* < 0.001). Levels of satisfaction were significantly higher among subjects who presented with controlled pressure figures (82.1 ± 12.1 SD vs. 77.5 ± 13.3 SD; *p* < 0.001).

## 4. Discussion

The sample of hypertensive study subjects showed a higher proportion of women than men, and a mean age of over 65 years. These characteristics are in consonance with the distribution of AHT, which affects more than 60% of people over the age of 60 years in developed countries [15]. The distribution of antihypertensive treatments, with a predominance of ACE inhibitors, ARA-II, and diuretics used in combination, is very similar to the findings reported by other studies conducted in Spain in recent years [16,17]; i.e., to the effect that up to one in three antihypertensive drugs corresponds to ACE inhibitors, and up to one in four corresponds to a diuretic or diuretics used in combination regimens. Our results showed a high proportion, as much as 19.0%, of participants who reported some adverse effect linked to their antihypertensive medication. Just under half the participants (48.3%) registered figures of 140/90 mmHg or higher, thus indicating that there is still a great potential for improvement, when it comes to the degree of control of blood pressure levels among hypertensive patients receiving pharmacological treatment. Adequate therapeutic adherence was achieved by three out of every four subjects studied. 

In a possible range of 0 through 100, participant treatment satisfaction scored a mean of almost 80 points, coinciding with the median. These results appear to indicate an acceptable level of satisfaction, which was higher among subjects who reported experiencing no adverse effects as a consequence of their antihypertensive medication. As was to be expected, levels of satisfaction were significantly lower among non-compliers with the treatment, and clearly higher among subjects who presented with well-controlled blood pressure levels. The score observed in the level of satisfaction with treatment is very similar to that obtained by the same questionnaire (SATMED-Q) in studies conducted on patients diagnosed with other diseases, such as transplant patients [10] or chronic disease sufferers [13] but is higher than that obtained among patients with insomnia [8] or diagnosed with rosacea [9].

By way of a limitation to the results obtained, non-responders or patients that do not go directly to a community pharmacy to obtain their medication—neither of whom are represented in our results—might well have a different level of satisfaction with their antihypertensive treatment when compared with patients who did participate in the study. The following were thus not represented: hypertensive subjects immobilized at home; and those affected by one or more exclusion criteria, such as patients with important mental deterioration or severe sensory deficits that rendered them incapable of undergoing the interviews required for the purposes of the study. Another study limitation could lie in a social desirability bias in the answers given by participants when questioned by health professionals. It should be noted too that, despite the MAQ being a validated test which displays a high specificity and high positive predictive value, using it to evaluate therapeutic adherence might have resulted in lack of adherence being underestimated due to reduced sensitivity. Furthermore, the use of non-probabilistic sampling to recruit participants might have given rise to a degree of selection bias. Lastly, since the study was carried out in only two community pharmacies in the same city, the external validity of the results may be moderate, and so the possibility of generalizing these to other populations with different healthcare or pharmaceutical care systems might thus be limited.

Although there are purpose-designed questionnaires for measuring treatment satisfaction in the case of some diseases, little attention has been paid to developing a more general measure of satisfaction capable of comparing different types of medication and different patient characteristics. A number of disease-specific treatment-satisfaction questionnaires have been developed, and studies have been conducted on different populations or groups of patients, e.g., persons diagnosed with gastro-esophageal reflux disease [18], HIV-infected patients [19], patients receiving anti-anemic treatment [20], and those diagnosed with idiopathic pulmonary fibrosis and atypical hemolytic uremic syndrome [21], Crohn’s disease [22], diabetes [23], osteoarthrosis [24], depressive disorders [25], etc.

Mention should be made of one generic questionnaire, the Treatment Satisfaction Questionnaire for Medication (TSQM), which measures four dimensions, namely, side effects, efficacy of treatment, ease of use, and general satisfaction. The TSQM is a measure of the main dimensions involved in patient satisfaction with medication and has been psychometrically validated in a heterogeneous sample [3,26]. It has been used on patients taking antihypertensive drugs, albeit in a shortened version that did not include questions on adverse effects in order to avoid interfering with patients’ behavior [27]. Even so, the TSQM has some limitations since it does not include aspects linked to satisfaction with healthcare or address how patients’ medication affects their daily life. These limitations encouraged the design of a new generic questionnaire, the Treatment Satisfaction with Medicines Questionnaire (SATMED-Q) [13], in which the above two aspects were included. 

Most treatment-satisfaction studies have used disease-specific questionnaires, but for more generalized use in daily practice, generic questionnaires would seem to be more suitable. To date, only the two generic questionnaires mentioned are suitable for evaluating patient satisfaction in any chronic disease. The SATMED-Q is a generic measure of patient-reported results for evaluating treatment satisfaction. Our results indicate that this questionnaire, validated in a sample of 484 hypertensive subjects, shows high internal consistency and good stability in the reliability analysis, yielding values higher than the accepted minimum standards [28]. Within its structure, there are six factors that are able to account for almost 90% of total variance, with the same dimensions being identified as those described in the original validation study [13], i.e., adverse effects of the antihypertensive medication, its impact on the performance of activities, ease and simplicity of administration, general satisfaction, and efficacy/effectiveness of the medication and physician follow-up. Previously, in a sample of 455 patients with chronic diseases [13], including type 2 diabetes, hypertension, osteoarthritis, benign prostatic hyperplasia, chronic obstructive pulmonary disease/asthma, depression, and migraine, the SATMED-Q also displayed high coherence and acceptable reliability. The authors concluded that it is a valid instrument for evaluating treatment satisfaction in chronic patients, though it would be desirable to study its capacity for ascertaining differences in satisfaction (sensitivity to change) in future prospective studies. 

The results obtained by Ruiz et al. [13], and those subsequently yielded by our own study show that the SATMED-Q is appropriate for use in routine clinical practice and in a community pharmacy or clinical research setting. Similarly, it can be employed both as a unidimensional instrument (using the total score), and for examining patient satisfaction with different aspects of treatment, since its component subscales can also be considered valid and reliable. From a feasibility standpoint, the response rate is highly satisfactory and the time of administration is very short, something that facilitates its use at any healthcare level, particularly in primary care or at community pharmacies, where the time available for attending to patients may well be limited. The results of this study suggest that the instrument enjoys very good acceptability among study subjects.

The validity of the questionnaire content was originally established by a panel of experts, and its construct validity by factor analysis, with the original dimensions being corroborated by our results. Another noteworthy aspect of the SATMED-Q is its generic nature, since the instrument can be used to compare patient satisfaction with pharmacological treatment, regardless of the type of drug or disease involved. Due to the low number of questionnaires that display this profile, this particular characteristic makes the instrument especially useful. The authors of the questionnaire have highlighted the need to conduct more studies which encompass other diseases and different drugs, in order to confirm the initial findings, as we did in our case, using a large sample of hypertensive subjects with any antihypertensive treatment guideline and any situation of comorbidity. 

Patient-centered care implies a change in attitude, inasmuch as it has to be accepted that the decision-making core is not exclusively anchored in the physician or other health professional [29]. It is becoming increasingly frequent for patients to be implicated in the choice of the best treatment for their disease and/or the decision about how to best approach their health problem. Such patient participation is directly related with the predominating medical culture: if this is paternalistic in nature, both physician and patient will assume that decision-making comes more properly within the exclusive responsibility of the former; in a more egalitarian cultural model; however, there is a tendency to opt for a formula of shared responsibility [30]. A good way of facilitating patient participation is to evaluate patient-perceived and -reported health outcomes objectively, accurately and with scientific rigor [31]. A treatment-satisfaction measure makes it possible to ascertain treatment-related aspects of most concern to the patient [32], thereby affording a genuine opportunity to improve the current treatment and consider these aspects for future treatments. The patient’s point of view is crucial, since there are multiple aspects, such as those related with ease of application and impact on daily activities, social interrelations and the like, which can solely be perceived by him/her [33].

## 5. Conclusions

In conclusion, the SATMED-Q, validated in a sample of 484 hypertensive subjects, shows high internal consistency and good stability in the reliability analysis. Its structure contains six discernible factors that can account for almost 90% of total variance. It is therefore an appropriate instrument for evaluating satisfaction with antihypertensive treatment, both in routine clinical practice and in community pharmacy or clinical research settings.

## Figures and Tables

**Figure 1 ijerph-18-03212-f001:**
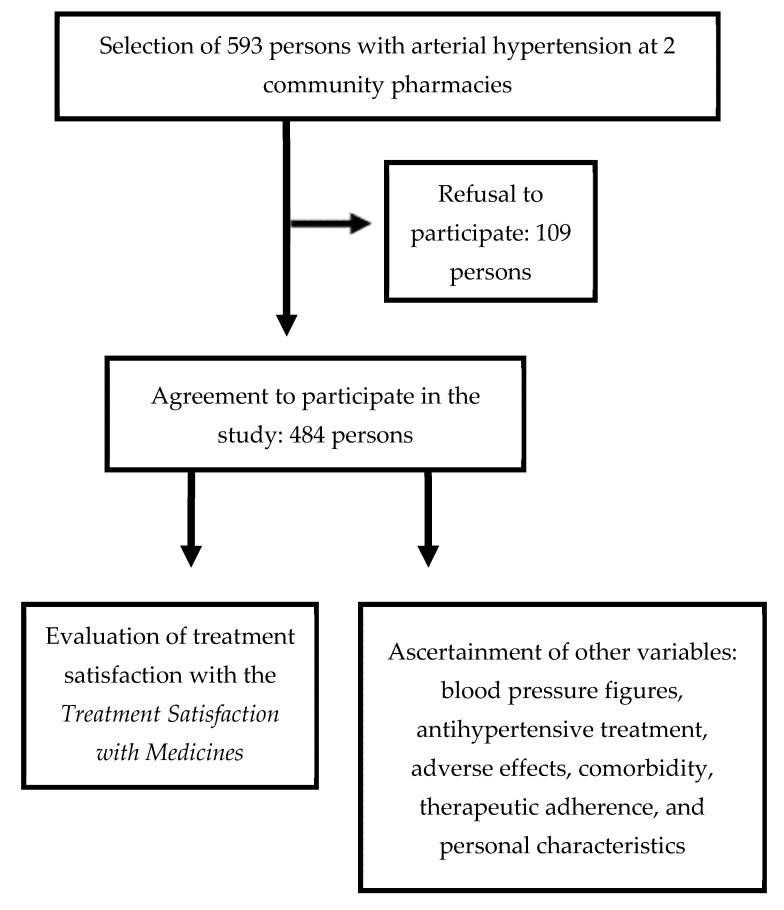
Study flow chart.

**Table 1 ijerph-18-03212-t001:** Evaluation of the homogeneity of the questionnaire (corrected item-total correlation).

Item		Item-Total Correlation
1	The side effects of the medication interfere in my physical activity	0.451
2	The side effects of the medication interfere in my leisure and spare-time activities	0.453
3	The side effects of the medication interfere in my daily activities	0.437
4	The medication that I’m taking relieves my symptoms	0.712
5	I’m satisfied with the time the medication takes before it starts to take effect	0.694
6	I feel better now than I did before taking the treatment	0.705
7	Taking my medication suits me fine	0.665
8	I find it easy to use/take the medication in its present form (taste, size, etc.)	0.646
9	The medication schedule suits me fine	0.667
10	Thanks to the medication that I’m taking, I’m better able to do my leisure and spare-time activities	0.800
11	Thanks to my medication, I’m better able to do my ablutions	0.771
12	Thanks to my medication I’m better able to do my daily activities	0.773
13	My physician has informed me in detail about my disease	0.625
14	My physician has told me how to treat my disease correctly	0.632
15	I intend to continue using this treatment	0.684
16	I’m happy with my treatment	0.750
17	In general, I’m satisfied with the treatment	0.766

**Table 2 ijerph-18-03212-t002:** Matrix of rotated components.

Items	Components					
	1	2	3	4	5	6
- The side effects of the medication interfere in my physical activity	0.948	0.070	0.068	0.099	0.078	0.070
- The side effects of the medication interfere in my leisure and spare-time activities	0.957	0.109	0.056	0.102	0.028	0.044
- The side effects of the medication interfere in my daily activities	0.956	0.074	0.061	0.069	0.030	0.026
- The medication that I’m taking relieves my symptoms	0.001	0.265	0.249	0.201	0.799	0.119
- I’m satisfied with the time the medication takes before it starts to take effect	0.061	0.178	0.241	0.188	0.844	0.101
- I feel better now than I did before taking the treatment	0.091	0.250	0.186	0.238	0.793	0.109
- Taking my medication suits me fine	0.049	0.160	0.898	0.193	0.186	0.095
- I find it easy to use/take the medication in its present form (taste, size, etc.)	0.075	0.127	0.895	0.167	0.215	0.078
- The medication schedule suits me fine	0.086	0.150	0.854	0.234	0.227	0.072
- Thanks to the medication that I’m taking, I’m better able to do my leisure and spare-time activities	0.140	0.854	0.181	0.221	0.274	0.119
- Thanks to my medication I’m better able to do my ablutions	0.088	0.882	0.150	0.194	0.201	0.165
- Thanks to my medication, I’m better able to do my daily activities	0.083	0.901	0.135	0.191	0.211	0.133
- My physician has informed me in detail about my disease	0.065	0.167	0.096	0.156	0.117	0.939
- My physician has told me how to treat my disease correctly	0.063	0.157	0.101	0.175	0.133	0.935
- I intend to continue using this treatment	0.090	0.160	0.267	0.800	0.207	0.117
- I’m happy with my treatment	0.111	0.233	0.191	0.868	0.199	0.164
- In general, I’m satisfied with the treatment	0.138	0.238	0.195	0.842	0.241	0.167

**Table 3 ijerph-18-03212-t003:** Total variance explained. Method of extraction: principal component analysis.

Component	Sum of Saturations to the Square of the Extraction			Sum of Saturations to the Square of the Rotation		
	Total	% Variance	% Accumulated	Total	% Variance	% Accumulated
1	7.474	43.967	43.967	2.838	16.692	16.692
2	2.530	14.884	58.851	2.759	16.232	32.924
3	1.713	10.077	68.928	2.742	16.127	49.051
4	7.474	8.276	77.204	2.558	15.049	64.100
5	1.141	6.710	83.915	2.451	14.418	78.518
6	1.029	6.051	89.965	1.946	11.448	89.965

**Table 4 ijerph-18-03212-t004:** Total variance explained in each factor in comparison with the results of the original study [13].

Component	% Variance	% Variance (Original Study)
Adverse effects of the antihypertensive medication	16.692	15.361
Impact of the medication on the performance of activities	16.232	35.591
Ease and simplicity of administration of the medication	16.127	10.449
General satisfaction with antihypertensive medication	15.049	6.144
Efficacy/effectiveness of antihypertensive medication	14.418	5.153
Physician follow-up	11.448	8.135

**Table 5 ijerph-18-03212-t005:** Loadings obtained by the items in the factor analysis.

Dimensions (Factors)	Loading	% Variance	% Accumulated
Adverse effects of the antihypertensive medication		16.692	16.692
- The side effects of the medication interfere in my physical activity	0.948		
- The side effects of the medication interfere in my leisure and spare time activities	0.957		
- The side effects of the medication interfere in my daily activities	0.956		
Impact of the medication on the performance of activities		16.232	32.924
- Thanks to the medication that I’m taking, I’m better able to do my leisure and spare time activities	0.854		
- Thanks to my medication, I’m better able to do my ablutions	0.882		
- Thanks to my medication, I’m better able to do my daily activities	0.901		
Ease and simplicity of administration of the medication		16.127	49.051
- Taking my medication suits me fine	0.898		
- I find it easy to use/take the medication in its present form (taste, size, etc.)	0.895		
- The medication schedule suits me fine	0.854		
General satisfaction with antihypertensive medication		15.049	64.100
- I intend to continue using this treatment	0.800		
- I’m happy with my treatment	0.868		
- In general, I’m satisfied with the treatment	0.842		
Efficacy/effectiveness of antihypertensive medication		14.418	78.518
- The medication that I’m taking relieves my symptoms	0.799		
- I’m satisfied with the time the medication takes before it starts to take effect	0.844		
- I feel better now than I did before taking the treatment	0.793		
Physician follow-up		11.448	89.965
- My physician has informed me in detail about my disease	0.939		
- My physician has told me how to treat my disease properly	0.935		

## Data Availability

Requestors wishing to access the trial data used in this study can make a request to joseba.rabanales@uclm.es.

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
