# Peer review of "Reliability and Validity of the Treatment Satisfaction with Medicines Questionnaire (SATMED-Q) in Persons with Arterial Hypertension"

_ijerph, 2021, doi:10.3390/ijerph18063212_

Round 1

Reviewer 1 Report

Please find my comments on the attached file

Author Response

This paper reflects the findings from pharmacists interviewing a large number of patients with hypertension from two community pharmacies from one district in one country where English is not the first language. Whilst there is a large sample and seemingly detailed statistics the results have limited generalisability, are open to social desirability bias (healthcare professionals asking patients face to face to report on how satisfied they are with the medicines they are providing them with) and the findings provide limited guidance with respect to practical application.

We agree that a possible limitation of the results lies in social desirability bias, which has now been mentioned in the limitations section.

I am not at all convinced by the need for such tools by the text provided within the introduction. Why don’t we just measure medication adherence directly or satisfaction with information provided rather than use a proxy measure. My other concern about such high level generic tools is that they just capture the problem and provide limited, if any insight, into what is required to address the underlying problem. Information covers a large number of bases and a lack of satisfaction with it does not tell the user what the underpinning problem actually is. The introduction needs to make a much more convincing case for the need for such a tool either in practice or for research purposes.

As suggested, the Introduction now argues in favor of the usefulness of these tools. It makes the point that it is important to have patients’ opinions about their medication, and one way of achieving this is to evaluate perceived outcomes objectively. Such patient-reported outcomes should include an evaluation of the treatment received without any interpretation of their responses by the physician, pharmacist or other health professional, and to ensure this, properly validated instruments are needed.

The authors use the term compliance throughout the article and then occasionally therapeutic adherence. Adherence is the accepted term as compliance suggest power and patients doing what they are told to do. This language requires correction throughout.

As suggested, the term “compliance” has been replaced by “therapeutic adherence” throughout the manuscript.

I had to read the article to understand what was meant by arterial hypertension as my original suspicion was that this was different to common primary hypertension. As I am not a cardiologist I don’t know the accepted international term but would like reassurance that this is the correct term.

It is indeed true to say that both “arterial hypertension” and “hypertension” are internationally accepted terms.

I am not a statistician but strongly suggest that this must be reviewed by a statistician with expertise in the concepts and techniques used within this paper. Whilst the authors demonstrate high internal consistency (with a tool which has already been validated) it may have been more useful to identify the minimum number of questions which could be used to obtain a final score which is still valid. In each construct there are three questions asking about the same topic and whilst this is good practise, it is of limited used with research to healthcare provision or for research purposes.

When delivering trials of any nature, participant burden is a major concern as it contributes to attrition and non-response. Consequently, short efficient questionnaires are preferred. These instruments are never used in isolation and consequently it is preferable to use instruments which have the minimum number of questions required.

While our aim here was to take a sample of persons with arterial hypertension and evaluate the reliability and validity of the SATMED-Q just as it was designed, we nonetheless feel that the reviewer’s comment is most apposite and that it would therefore be of great interest to ascertain the validity of a shorter and more efficient questionnaire in a future study.

As hypertension is largely symptomless, I cannot see how its control is associated with ability to do ablutions. Furthermore, the dosing frequency is usually once a day for most hypertension treatments and again I cannot see the value of the question regarding dosing frequency. Consequently I believe such questions are extraneous in this population.

The SATMED-Q was designed to be used on chronic patients undergoing pharmacological treatment for any disease, and it is thus possible that some items might be more appropriate than others in some specific diseases. However, regardless of whether it is a disease with more or fewer symptoms, we should also consider the possible adverse effects of the treatments and their possible repercussions on any activity.

The authors introduce the TSQM in the discussion when it should be introduced in the introduction and the argument made for the need for this research made at that point.

As suggested, this questionnaire is now also cited in the Introduction.

The authors report that they received permission to use the main tool but do not state the same for Morisky. Morisky has given the rights to his tools to an individual who sues anyone who uses the tool without permission. Consequently, the authors need to ensure that they had permission to use the tool.

The Morisky questionnaire used was the MAQ (Medication Adherence Questionnaire), which is in the public domain and corresponds to reference no. 14 (Morisky, D.E.; Green, L.W.; Levine, D.M. Concurrent and predictive validity of a self-reported measure of medication adherence. Med. Care. 1986, 24, 67-74. doi: 10.1097/00005650-198601000-00007). The Morisky questionnaires that are copyright-protected are the MMAS-4 and MMAS-8, and are different.

The Morisky tools themselves are very poor predictors of adherence with low sensitivity, consequently this limitation requires reporting. I suspect both tools are the same with respect to medicines use i.e. Patients are largely unwilling to admit to perceived deviant behaviours. At least Morisky is usually completed by the patient and not via interview.

In the Discussion, mention is now made of the lack of sensitivity of the questionnaire used.

The blood pressure was measured twice but was this consecutively in the same visit or separated (as per guidelines)? What did the authors do to ensure that it was valid i.e. how much time were the individuals given to sit and relax before the BP was measured?

We have now specified in Materials and Methods that the measurements were made on 2 occasions, after a minimum five-minute rest period, with the subject in a seated position, his/her spine firmly supported by the back of the chair and arm in a semi-flexed position at the level of the heart.

I am never sure why such tools are correlated with demographic factors which cannot be changed and therefore knowing the relationship is of limited practical use. The authors state that they excluded individuals with low intellectual performance but then report that a significant proportion of participants had low levels of literacy. The method requires greater clarification with respect to this exclusion criteria.

The Materials and Methods section explains that persons with low intellectual performance were excluded, since they presented with a degree of cognitive impairment incompatible with undergoing an interview. Even so, low educational level did not prove to be a bar to obtaining data.

Sampling was described as consecutive non-probability which suggests that patients were approached by the pharmacists for inclusion. Were all patients eligible to participate approached or was this done at the convenience of the pharmacists perhaps with a distinct opportunity for selection bias i.e. they chose the patients they felt most comfortable asking. In summary, I believe there are a significant number of limitations within this study which require reporting in the discussion and perhaps more up front in the discussion rather than at the end i.e. the results need to be considered in light of the following limitations. Seeking feedback on satisfaction with care by the care providers themselves through interview is a major limitation.

Despite using non-probabilistic sampling, all patients who fulfilled the selection criteria were included in the study. That said, however, the use of non-probabilistic sampling and the possibility of having incurred some selection bias have also been listed as a limitation. The limitations section has been moved to the beginning of the Discussion.

The authors then need to be more careful with respect to their recommendations. This tool cannot be recommended for international use in this population based on this one limited study which probably only represents the experiences of patients within a limited health system, a relatively small number of medical practitioners and pharmacists.

Another limitation now flagged by us is the moderate external validity of the results and the limited possibilities of generalizing these to other populations with different healthcare or pharmaceutical care systems.

The language needs careful consideration for an international audience and in places sentences have errors within them.

Our apologies for this: we have now made every effort to detect and correct any possible errors in the text.

A statistician with appropriate expertise needs to review this paper and it would be of much greater use or interest if the authors identified the small number of questions which could be used effectively in combination to assess satisfaction in this population.

We are most grateful for these suggestions. For our part, we have taken the statistical advice of a suitably qualified person. In a future analysis, we will seek to provide a shorter version of the questionnaire.

Reviewer 2 Report

 1. It´s necessary to introduce more aspects of patient satisfaction that influence adherence to treatment, for example: quality of care, information about treatment, information transmitted by the doctor or other professional.

  2. Table 2 is difficult to read in that format.

  3. Discussion, it´s necessary to compare their results with other studies that use similar questionnaires in other diseases, as stated by the authors. It is a very general discussion. 

Author Response

  1. It´s necessary to introduce more aspects of patient satisfaction that influence adherence to treatment, for example: quality of care, information about treatment, information transmitted by the doctor or other professional.

On addressing the topic of therapeutic adherence in the Introduction, we make reference to these very aspects.

  1. Table 2 is difficult to read in that format.

In the Table, the numbers have now been spaced farther apart for ease of reading.

  1. Discussion, it´s necessary to compare their results with other studies that use similar questionnaires in other diseases, as stated by the authors. It is a very general discussion.

In the Discussion, we have now included a comparison with the results obtained in 4 studies in which the same questionnaire was used to evaluate satisfaction in other diseases, such as insomnia, rosacea, transplant patients and chronic patients (diabetes, osteoarthrosis, etc.).

Reviewer 3 Report

In general, the article is clear and of interest, although some recommendations can be made to the authors:

- I think the Citation on the first page, on the left, is not correct.
- In the summary (lines 20-21), "participants’ characteristics "are mentioned. In 2. Materials and Methods is mentioned that: "the characteristics of the study subjects are described with greater detail in a previously published paper (line 84)". It would be interesting to insert these characteristics here and without having to go to the previous article. They seem important enough in this work. Then they appear on line 117 (age, sex and educational level).
- A dot is missing after the parentheses (line 29).
- The 2. Materials and Methods describes that the sample is interviewed in two pharmacies in a city in which there are more than 200 pharmacies. Are these pharmacies representative of the population? This regardless of whether the sample size is correct (line 99). This question is linked to the previous comment. What area the participants characteristics?
- In section 2 it is not clear how the "possible reported adverse effects" will be measured or valued (lines 116 and 149). It is clearer on line 165. Perhaps some information befor should be useful.
- The data for "feasibility determined by reference to the percentage of unanswered questions" (lines 119-120) does not appear in the results. It must be an interesting data.
- Some reference is missed in the section Limitations, or in the section of methodology, to what are the characteristics of the people who refuse to participate or the reasons they have (line 99). They are 18.3%.
- In line 330, are intentional the quotes after Contributions:?

Author Response

In general, the article is clear and of interest, although some recommendations can be made to the authors:

- I think the Citation on the first page, on the left, is not correct.

As suggested, the citation has now been corrected.

- In the summary (lines 20-21), "participants’ characteristics "are mentioned. In 2. Materials and Methods is mentioned that: "the characteristics of the study subjects are described with greater detail in a previously published paper (line 84)". It would be interesting to insert these characteristics here and without having to go to the previous article. They seem important enough in this work. Then they appear on line 117 (age, sex and educational level).

Information has now been added on the proportion of persons who lived alone or were institutionalized. Data has likewise been added regarding the occupational status of the participants.

- A dot is missing after the parentheses (line 29).

This has now been corrected.

- The 2. Materials and Methods describes that the sample is interviewed in two pharmacies in a city in which there are more than 200 pharmacies. Are these pharmacies representative of the population? This regardless of whether the sample size is correct (line 99). This question is linked to the previous comment. What area the participants characteristics?

There are currently 84 pharmacies in the city of Albacete. The two pharmacies at which the participants were interviewed are respectively situated in the center and on the outskirts of the city, with these two areas having different socioeconomic levels. In our opinion, the persons interviewed thus constitute a representative sample of the hypertensive population.

- In section 2 it is not clear how the "possible reported adverse effects" will be measured or valued (lines 116 and 149). It is clearer on line 165. Perhaps some information befor should be useful.

We have now specified in Materials and Methods that these are adverse effects linked to antihypertensive medication.

- The data for "feasibility determined by reference to the percentage of unanswered questions" (lines 119-120) does not appear in the results. It must be an interesting data.

By virtue of being a hetero-administered questionnaire, responses could be obtained to all of the items in the SATMED-Q questionnaire, something that is now mentioned under Results.

- Some reference is missed in the Limitations section, or in the section of methodology, to what are the characteristics of the people who refuse to participate or the reasons they have (line 99). They are 18.3%.

We have now specified that most persons who refused to participate did so because they did not have the time to undergo the interview, and in a small number of cases because of health reasons.

- In line 330, are intentional the quotes after Contributions?

These quotes have now been removed.